# STOCHASTIC REWEIGHTED GRADIENT DESCENT

## ABSTRACT

Importance sampling is a promising strategy for improving the convergence rate of stochastic gradient methods. It is typically used to precondition the optimization problem, but it can also be used to reduce the variance of the gradient estimator. Unfortunately, this latter point of view has yet to lead to practical methods that provably improve the asymptotic error of stochastic gradient methods. In this work, we propose stochastic reweighted gradient (SRG), a variance-reduced stochastic gradient method based solely on importance sampling that can improve on the asymptotic error of stochastic gradient descent (SGD) in the strongly convex and smooth case. We show that SRG can be extended to combine the benefits of both importance-sampling-based preconditioning and variance reduction. When compared to SGD, the resulting algorithm can simultaneously reduce the condition number and the asymptotic error, both by up to a factor equal to the number of component functions. We demonstrate improved convergence in practice on $\ell_2$-regularized logistic regression problems.

## 1 INTRODUCTION

Unconstrained optimization of finite-sum objectives is a core algorithmic problem in machine learning. The prototypical way of solving such problems is by viewing them through the lens of stochastic optimization, where the source of stochasticity resides in the choice of the index in the sum. Stochastic gradient descent (SGD) (Robbins & Monro, 1951) remains the standard algorithm for this class of problems.

A natural way to improve on SGD is by considering importance sampling schemes. This idea is not new and dates back to (Needell et al., 2014) who uses importance sampling as a preconditioning technique. They propose sampling the indices with probabilities proportional to the smoothness constants of the corresponding component functions, and show that this sampling scheme provably reduces the condition number of the problem.

In another line of work, variance-reduced methods were found to achieve linear convergence in the strongly convex and smooth case (Roux et al., 2012; Schmidt et al., 2017). Many of these methods rely on control variates to reduce the variance of the gradient estimator used by SGD (Johnson & Zhang, 2013; Defazio et al., 2014). While very successful, the applicability of these methods is limited by the large memory overhead that they introduce, or the periodic full-gradient recomputation that they require (Defazio & Bottou, 2019). Despite strong progress in this research area, an importance-sampling-based analogue to these algorithms, which is potentially free from these drawbacks, has yet to emerge.

In this work, we propose such an analogue. We introduce stochastic reweighted gradient (SRG), an importance-sampling-based variance-reduced optimization algorithm. Similar to SGD, SRG requires a single gradient oracle call per iteration, and only requires $O(n)$ additional memory, and $O(\log n)$ additional floating point operations per iteration. We analyze the convergence rate of SRG in the strongly-convex and smooth case, and show that it can provably improve the asymptotic error of SGD. Finally, we show how our importance sampling strategy can be combined with smoothness-based importance sampling, and prove that the resulting algorithm simultaneously performs variance reduction and preconditioning. We demonstrate improved convergence in practice on $\ell_2$-regularized logistic regression problems.

## 2 PRELIMINARIES

We consider the finite-sum optimization problem:

$$\min_{x \in \mathbb{R}^d} \left\{ F(x) := \frac{1}{n} \sum_{i=1}^{n} f_i(x) \right\} \qquad (1)$$

where $F$ is $\mu$-strongly convex for $\mu > 0$, and for all $i \in [n]$, $f_i$ is convex and $L_i$-smooth for $L_i > 0$. Note that by strong-convexity, $F$ has a unique minimizer $x^* \in \mathbb{R}^d$. We define the maximum $L_{\max} := \max_{i \in [n]} L_i$ and average $\overline{L} := \sum_{j=1}^{n} L_i/n$ smoothness constants. Similarly, we define the maximum $\kappa_{\max} := L_{\max}/\mu$ and average $\overline{\kappa} := \overline{L}/\mu$ condition numbers.

The classical way of solving (1) is by viewing it as a stochastic optimization problem where the randomness comes from the choice of the index $i \in [n]$. Starting from some $x_0 \in \mathbb{R}^d$, and for an iteration number $k \in \mathbb{N}$, stochastic gradient descent (SGD) performs the following update:

$$x_{k+1} = x_k - \alpha_k \nabla f_{i_k}(x_k)$$

for a step size $\alpha_k > 0$ and a random index $i_k$ drawn uniformly from $[n]$.

The idea behind importance sampling for SGD is to instead sample the index $i_k$ according to a chosen distribution $p_k$ on $[n]$, and to perform the update (Needell et al., 2014):

$$x_{k+1} = x_k - \alpha_k \frac{1}{np_k^{i_k}} \nabla f_{i_k}(x_k) \qquad (2)$$

where $p_k^i$ is the $i^{th}$ component of the probability vector $p_k$. It is immediate to verify that the importance sampling estimator of the gradient is unbiased as long as $p_k > 0$.

The question that we address in this paper is how to design a sequence $\{p_k\}_{k=0}^{\infty}$ that produces more efficient gradient estimators than the ones produced by uniform sampling. One way to design such a sequence is by adopting a greedy strategy: at each iteration $k$ we choose $p_k$ to minimize the conditional variance of the gradient estimator, which is given by, up to an additive constant:

$$\sigma^2(x_k, p) := \mathbb{E}_{i_k \sim p} \left[ \left\| \frac{1}{np^{i_k}} \nabla f_{i_k}(x_k) \right\|_2^2 \, \middle| \, (i_t)_{t=0}^{k-1} \right] = \frac{1}{n^2} \sum_{i=1}^{n} \frac{1}{p^i} \left\| \nabla f_i(x_k) \right\|_2^2, \qquad (3)$$

This conditional variance is minimized at (Zhao & Zhang, 2015):

$$\arg\min_{p \in \Delta} \sigma^2(x_k, p) = \left( \frac{\left\| \nabla f_i(x_k) \right\|_2}{\sum_{j=1}^{n} \left\| \nabla f_j(x_k) \right\|_2} \right)_{i=1}^{n} \qquad (4)$$

Ideally, we would like to set $p_k$ to this minimizer. However, this requires knowledge of the gradient norms $(\left\| \nabla f_i(x_k) \right\|_2)_{i=1}^{n}$, which, in general, requires $n$ gradient evaluations per iteration.

## 3 ALGORITHM

In this section, we show how to design a tractable sequence of importance sampling distributions for SGD that approximate the conditional-variance-minimizing distributions (4). First we construct efficient approximations of the conditional variances (3). We then state a simple bound on the approximation errors and use it to motivate our choice of importance sampling distributions.

To approximate the conditional variances, we follow the strategy of certain variance-reduced methods (Roux et al., 2012; Schmidt et al., 2017; Defazio et al., 2014). These methods maintain a table $(g_k^i)_{i=1}^{n}$ that tracks the component gradients $(\nabla f_i(x_k))_{i=1}^{n}$ and updates itself each iteration at the index $i_k$ used to update the iterates $x_k$ (2). Our method instead maintains an array of gradient *norms*, from which we construct an approximation of the conditional variance (3) of the gradient estimator:

$$\tilde{\sigma}^2(x_k, p) := \frac{1}{n^2} \sum_{i=1}^{n} \frac{1}{p^i} \left\| g_k^i \right\|_2^2$$

---

**Algorithm 1** SRG

---

1: **Parameters:** step sizes $(\alpha_k)_{k=0}^{\infty} > 0$, mixture coefficients $(\theta_k)_{k=0}^{\infty} \in (0, 1]$
2: **Initialization:** $x_0 \in \mathbb{R}^d, (\|g_0^i\|_2)_{i=1}^n \in \mathbb{R}^n$
3: **for** $k = 0, 1, 2, \ldots$ **do**
4:     $p_k = (1 - \theta_k)q_k + \theta_k/n$                                                $\{q_k$ is defined in (5)$\}$
5:     $b_k \sim \text{Bernoulli}(\theta_k)$
6:     **if** $b_k = 1$ **then** $i_k \sim 1/n$ **else** $i_k \sim q_k$
7:     $x_{k+1} = x_k - \alpha_k \frac{1}{np_k^{i_k}} \nabla f_{i_k}(x_k)$
8:     $\|g_{k+1}^i\|_2 = \begin{cases} \|\nabla f_i(x_k)\|_2 & \text{if } b_k = 1 \text{ and } i_k = i \\ \|g_k^i\|_2 & \text{otherwise} \end{cases}$
9: **end for**

---

which is minimized at:

$$q_k = \operatorname*{arg\,min}_{p \in \Delta} \tilde{\sigma}^2(x_k, p) = \left( \frac{\|g_k^i\|_2}{\sum_{j=1}^n \|g_k^j\|_2} \right)_{i=1}^n \tag{5}$$

We do not directly use $q_k$ as an importance sampling distribution, because this approximation may be poor. In particular, we have the following bound on the conditional variance (3):

$$\sigma^2(x_k, p) \leq \frac{2}{n^2} \sum_{i=1}^n \frac{1}{p^i} \|\nabla f_i(x_k) - g_k^i\|_2^2 + 2\tilde{\sigma}^2(x_k, p) \tag{6}$$

Recall that our goal is to pick $p_k$ that minimizes $\sigma^2(x_k, p)$. $q_k$ minimizes the second term on the right-hand side, but we must ensure that both terms are small. Two conditions are needed to keep the first term small. The first is to control the terms $\|\nabla f_i(x_k) - g_k^i\|_2^2$, which we can do by making sure that the historical gradients $g_k^i$ are frequently updated. The second is to ensure that the probabilities $p_k^i$ are lower bounded. We achieve both of these properties, as well as approximately minimize the right-hand side of (6) by mixing $q_k$ with the uniform distribution over $[n]$. This yields our final importance sampling distribution for a given mixture coefficient $\theta_k \in (0, 1]$:

$$p_k = (1 - \theta_k)q_k + \frac{\theta_k}{n} \tag{7}$$

Our analysis clarifies the role of the sequence of mixture coefficients $(\theta_k)_{k=0}^{\infty}$, and relates it to both the step size sequence $(\alpha_k)_{k=0}^{\infty}$ and the asymptotic error of SRG in the constant step size setting.

Curiously, our analysis requires performing the update of the array $(\|g_k^i\|_2)_{i=1}^n$ only when the index $i_k$ is drawn from the uniform mixture component. It is not clear to us whether this constraint is an artifact of the analysis or a property of the algorithm. We discuss this further after the statement of Lemma 1 in section 4. The pseudocode for our method is given in Algorithm 1.

We briefly discuss how SRG can be efficiently implemented. To sample from $q_k$, we store $(\|g_k^i\|_2)_{i=1}^n$ in a binary indexed tree using $O(n)$ memory, along with the normalizing constant $\lambda_k = \sum_{i=1}^n \|g_k^i\|_2$. The binary indexed tree can be updated and maintained in $O(\log n)$ operations, while the normalizing constant can be maintained in constant time. We can then sample from $q_k$ using inverse transform sampling: we multiply a uniform random variable $u \in [0, 1)$ by $\lambda_k$, then return the largest index whose corresponding prefix sum is less than $\lambda_k u$. This procedure also requires only $O(\log n)$ operations. The total overhead of SRG when compared to SGD is therefore $O(n)$ memory and $O(\log n)$ floating point operations per iteration.

## 4 THEORY

In this section, we analyze the convergence rate of SRG, and show that it can achieve a better asymptotic error than SGD. Two key constants are helpful in contrasting the asymptotic errors of

SRG and SGD. Recall the definition of $\sigma^2(x_k, p_k)$ in (3), and define:

$$\sigma^2 := \sigma^2(x^*, 1/n) = \frac{1}{n} \sum_{i=1}^{n} \|\nabla f_i(x^*)\|_2^2$$

$$\sigma_*^2 := \min_{p \in \Delta} \sigma^2(x^*, p) = \frac{1}{n^2} \left( \sum_{i=1}^{n} \|\nabla f_i(x^*)\|_2 \right)^2$$

It is well known that the asymptotic error of SGD depends linearly on $\sigma^2$ (Needell et al., 2014). We here show that SRG reduces this to a linear dependence on $\sigma_*^2$, which can be up to $n$ times smaller.

To study the convergence rate of SRG, we use the following Lyapunov function, which is similar to the one used to study the convergence rate of SAGA (Hofmann et al., 2015; Defazio, 2016):

$$T^k = T(x_k, (g_k^i)_{i=1}^n) := \frac{\alpha_k}{\theta_k} \frac{a}{L_{\max}} \sum_{i=1}^{n} \|g_k^i - \nabla f_i(x^*)\|_2^2 + \|x_k - x^*\|_2^2 \tag{8}$$

for a constant $a > 0$ that we set during the analysis. The proofs of this section are in Appendix B.

### 4.1 Intermediate lemmas

Before proceeding with the main result, let us first state two intermediate lemmas. The first studies the evolution of $(g_k^i)_{i=1}^n$ from one iteration to the next.

**Lemma 1.** *Let $k \in \mathbb{N}$ and suppose that $(g_k^i)_{i=1}^n$ evolves as in Algorithm 1. Taking expectation with respect to $(b_k, i_k)$, conditional on $(b_t, i_t)_{t=0}^{k-1}$, we have:*

$$\mathbb{E}\left[ \sum_{i=1}^{n} \|g_{k+1}^i - \nabla f_i(x^*)\|_2^2 \right] \leq \left(1 - \frac{\theta_k}{n}\right) \sum_{i=1}^{n} \|g_k^i - \nabla f_i(x^*)\|_2^2 + 2\theta_k L_{max} [F(x_k) - F(x^*)]$$

The use of the Bernoulli random variable $b_k$ in Algorithm 1 to monitor the update of $(g_k^i)_{i=1}^n$ is necessary for Lemma 1 to hold. In particular, without the use of $b_k$, the elements of $(g_k^i)_{i=1}^n$ may have different probabilities of being updated. In that case, the second term of Lemma 1 becomes a weighted average of terms that are technically difficult to deal with. When the importance sampling distribution does not depend on the iteration, this issue can be fixed with a slight modification of the Lyapunov function (8) (Schmidt et al., 2015). In our case however, we are dealing with time-varying importance sampling distributions, and this approach fails. This is why we rely on the Bernoulli random variable $b_k$ to ensure that the probability of updating any $g_k^i$ is fixed to $\theta_k/n$.

The second lemma is a bound on the conditional variance of the gradient estimator used by SRG. We intentionally leave as many free parameters as possible in the bound and optimize over them in the main result of section 4.2.

**Lemma 2.** *Let $k \in \mathbb{N}$ and assume that $\theta_k \in (0, 1/2]$. Taking expectation with respect to $(b_k, i_k)$, conditional on $(b_t, i_t)_{t=0}^{k-1}$, we have, for all $\beta, \gamma, \delta, \eta > 0$:*

$$\mathbb{E}_{i_k \sim p_k} \left[ \left\| \frac{1}{n p_k^{i_k}} \nabla f_{i_k}(x_k) \right\|_2^2 \right] \leq \frac{2D_1 L_{max}}{\theta_k} [F(x_k) - F^*] + \frac{D_2}{\theta_k n} \sum_{i=1}^{n} \|g_k^i - \nabla f_i(x^*)\|_2^2 + D_3 (1 + 2\theta_k) \sigma_*^2$$

*where $D_1, D_2$ and $D_3$ are positive functions of the free parameters $\beta, \gamma, \delta, \eta$.*

### 4.2 Main result

Our main result is a bound on the evolution of the Lyapunov function $T^k$ along the steps of SRG.

**Theorem 1.** *Suppose that $(x_k, (g_k^i)_{i=1}^n)$ evolves according to Algorithm 1. Further, assume that for all $k \in \mathbb{N}$: (i) $\alpha_k/\theta_k$ is non-increasing. (ii) $\theta_k \in (0, 1/2]$. (iii) $\alpha_k \leq \theta_k/12L_{max}$. Then:*

$$\mathbb{E}\left[ T^{k+1} \right] \leq (1 - \rho_k) \mathbb{E}\left[ T^k \right] + (1 + 2\theta_k) 6\alpha_k^2 \sigma_*^2$$

*for all $k \in \mathbb{N}$, and where:*

$$\rho_k := \min\left\{\frac{\theta_k}{12n}, \alpha_k\mu\right\}$$

The constants ($1/12$ in the bound on the step size and in $\rho_k$, 6 in front of the $\sigma_*^2$ term) in this theorem are optimized under the following constraints. First, the parameterized form of the above bound shows that the largest allowable step size is given by $c_2\theta_k/L_{\max}$. Using it we get:

$$\rho_k = \min\left\{c_1\frac{\theta_k}{n}, c_2\frac{\theta_k}{\kappa_{\max}}\right\}$$

for some constants $c_1, c_2 > 0$. As we do not know a priori the relative magnitudes of $n$ and $\kappa_{\max}$, and since $c_1$ and $c_2$ are inversely proportional, we impose the constraint $c_1 = c_2$. Similarly, our parameterized bound gives an asymptotic error of the form $(1 + 2\theta_k)c_3\alpha_k^2\sigma_*^2$ for a constant $c_3$. We chose to impose the constraint $c_3 = 6$. $c_1, c_2$ and $c_3$ are all functions of the free parameters of Lemma 2 and the constant $a$ of the Lyapunov function (8). Numerically maximizing $c_2$ (and therefore the largest allowable step size) subject to these two constraints ($c_3 = 6$ and $c_1 = c_2$) with respect to these parameters yields the result in Theorem 1.

To give the reader an idea of the sensitivity of the result to the choice of $c_3$, note that setting $c_3 = 2$ yields $c_2 \approx 1/20$, whereas taking $c_3 \gg 1$ yields $c_2 \approx 1/10$. We have attempted to obtain the best bound possible on the largest step size allowable, but the rather small prefactor $c_2$ seems to be an inevitable consequence of the multiple (but as far as we can tell necessary) uses of Young's inequality in our analysis. Our experiments suggest that the dependence on the mixture coefficient $\theta$ is real, but that the prefactor $c_2 = 1/12$ may be an artifact of the analysis.

For SRG with a constant mixture coefficient and step size, its convergence rate and complexity are characterized by the following corollary of Theorem 1:

**Corollary 1.** *Suppose that $(x_k, (g_k^i)_{i=1}^n)$ evolves according to Algorithm 1 with a constant mixture coefficient $\theta_k = \theta \in (0, 1/2]$ and a constant step size $\alpha_k = \alpha \le \theta/12L_{max}$. Then for any $k \in \mathbb{N}$:*

$$\mathbb{E}\left[T^k\right] \le (1-\rho)^k T^0 + (1+2\theta)\frac{6\alpha^2\sigma_*^2}{\rho}$$

*where $\rho = \rho_k$ is as defined in Theorem 1. For any $\varepsilon > 0$ and $\theta \in (0, 1/2]$, choosing:*

$$\alpha = \min\left\{\frac{\theta}{12L_{max}}, \frac{\varepsilon\mu}{(1+2\theta)12\sigma_*^2}, \sqrt{\frac{\theta}{1+2\theta}\frac{\varepsilon}{144n\sigma_*^2}}\right\}$$

*and:*

$$k \ge \max\left\{\frac{12n}{\theta}, \frac{1}{\alpha\mu}\right\}\log\left(\frac{2T^0}{\varepsilon}\right)$$

*guarantees $\mathbb{E}\left[\|x_k - x^*\|_2^2\right] \le \varepsilon$*

Comparing the convergence rate of SRG in Corollary 1 with the standard result for SGD (Needell et al., 2014), we see that they are of similar form. When $\rho = \alpha\mu$, the bound of Corollary 1 is better asymptotically. Indeed, in this case, and as $k \to \infty$, the iterates of SRG stay within a ball of radius $O(\sqrt{\sigma_*^2})$ of the minimizer, while those of SGD stay withing a ball of radius $O(\sqrt{\sigma^2})$. The equality $\rho = \alpha\mu$ holds when $\alpha \le 1/n\mu$, which is true for all allowable step sizes if the problem is dominated by its maximum condition number $\kappa_{\max} \ge n$, and for small step sizes otherwise.

In terms of complexity, we have the following comparison. Up to constants, the complexity of SRG with a constant mixture coefficient and step size is of the form:

$$O\left(n + \sqrt{\frac{n\sigma_*^2}{\mu^2\varepsilon}} + \kappa_{\max} + \frac{\sigma_*^2}{\mu^2\varepsilon}\right)\log\left(\frac{1}{\varepsilon}\right)$$

We compare this to the complexity of SGD with constant step size (Needell et al., 2014):

$$O\left(\kappa_{\max} + \frac{\sigma^2}{\mu^2\varepsilon}\right)\log\left(\frac{1}{\varepsilon}\right)$$

In the high accuracy regime, the $\varepsilon^{-1}$ terms dominate the complexities of SRG and SGD. In this case, SRG enjoys a better complexity than SGD since $\sigma_*^2 \le \sigma^2$.

---

**Algorithm 2** SRG+

---

**Parameters:** step sizes $(\alpha_k)_{k=0}^{\infty} > 0$, mixture coefficients $(\theta_k)_{k=0}^{\infty} \in (0, 1]$
**Initialization:** $x_0 \in \mathbb{R}^d$, $(\lVert g_0^i \rVert_2)_{i=1}^n \in \mathbb{R}^n$
**for** $k = 0, 1, 2, \ldots$ **do**
    $p_k = (1 - \theta_k)q_k + \theta_k v$                    $\{q_k$ is defined in (5), $v$ is defined in (11)$\}$
    $b_k \sim \text{Bernoulli}(\theta_k)$
    **if** $b_k = 1$ **then** $(i_k, j_k) \sim \pi$ **else** $i_k \sim q_k$         $\{\pi$ maximally couples $(v, 1/n)\}$
    $x_{k+1} = x_k - \alpha_k \frac{1}{np_k^{i_k}} \nabla f_{i_k}(x_k)$
    $\left\lVert g_{k+1}^j \right\rVert_2 = \begin{cases} \lVert \nabla f_j(x_k) \rVert_2 & \text{if } b_k = 1 \text{ and } j = j_k \\ \left\lVert g_k^j \right\rVert_2 & \text{otherwise} \end{cases}$
**end for**

---

## 5 EXTENSION

In this section, we extend SRG to combine its variance reduction capacity with the preconditioning ability of smoothness-based importance sampling.

A straightforward way to generalize the argument given in the derivation of SRG in section 3 is to consider the following bound on the conditional variance (3) of the gradient estimator, which can be derived from Young's inequality and the $L_i$-smoothness of each $f_i$:

$$\sigma^2(x_k, p) \leq \frac{3}{n^2} \sum_{i=1}^n \frac{L_i}{p^i} \langle \nabla f_i(x_k) - \nabla f_i(x^*), x_k - x^* \rangle + \frac{3}{n^2} \sum_{i=1}^n \frac{1}{p^i} \left\lVert g_k^i - \nabla f_i(x^*) \right\rVert_2^2 + 3\tilde{\sigma}^2(x_k, p) \tag{9}$$

While the motivating bound (6) seems more intuitive, because we think of $g_k^i$ as tracking $\nabla f_i(x_k)$, it turns out that this second bound better captures the evolution of $g_k^i$ in relation to $\nabla f_i(x_k)$. At a high-level, this is because $g_k^i$ tracks $\nabla f_i(x_k)$ indirectly: both hover around $\nabla f_i(x^*)$ as $k$ gets large.

Similar to our approach in section 3, our goal is to pick $p_k$ that minimizes the right-hand side of (9). We know that $q_k$ (5) minimizes the third term, but we need to make sure that the first two are also small. To minimize the first term, knowing nothing about the relative sizes of the inner product terms, it makes sense to have probabilities proportional to the smoothness constants [1]. On the other hand, to keep the second term small, we need to ensure that the historical gradients $(g_k^i)_{i=1}^n$ are frequently updated, which we can do by imposing a uniform lower bound on the probabilities. These considerations motivate us to consider the following distributions:

$$p_k' = (1 - \eta_k - \theta_k)q_k + \eta_k v + \frac{\theta_k}{n} \tag{10}$$

for positive mixture coefficients $(\theta_k, \eta_k)$ satisfying $\theta_k + \eta_k \in (0, 1]$, and where $v$ is given by:

$$v = \left( \frac{L_i}{n\overline{L}} \right)_{i=1}^n \tag{11}$$

Using these probabilities, we are able to show that the resulting algorithm does indeed achieve both variance reduction and preconditioning. However, in the worst case, its complexity is twice as much as what we would expect from simply replacing $L_{\max}$ with $\overline{L}$ in Corollary 1. Intuitively, this is because the probability assigned to the uniform component in (7) needs to be split between the uniform and the smoothness-based components in (10).

### 5.1 CAREFULLY DECOUPLING THE UPDATES OF $(g_k^i)_{i=1}^n$ AND $x_k$

Here we show how to design our algorithm such that this additional factor of 2 (described above) in the complexity is reduced to:

$$1 + \lVert v - 1/n \rVert_{TV} \leq 2 - 1/n$$

---

[1]Based on this argument alone, one would want them to be proportional to the square root of the smoothness constants. This however does not lead to a nice averaging of the inner product terms, which is important for technical reasons related to the strong-convexity of $F$ but not of the component functions $f_i$.

Following (Schmidt et al., 2015), our method is based on the observation that we can decouple the index used to update the historical gradients and the index used to update the iterates, which we refer to as $j_k$ and $i_k$, respectively. Intuitively, to minimize (10) we would ideally want $j_k$ to be uniformly distributed and $i_k$ to be distributed according to $p_k = (1 - \theta_k)q_k + \theta_k v$. Unfortunately, sampling $(j_k, i_k)$ independently with these marginals does not address our issue, because we would still require an average of approximately two gradient evaluations per iteration.

Luckily, we can use any coupling between $(j_k, i_k)$, because the evolution of the Lyapunov function (8) only depends on the marginal distributions and not the joint. In particular, we obtain the same bound on the evolution of $T^k$ regardless of how $i_k$ and $j_k$ are coupled. It therefore makes sense to pick the coupling that maximizes the probability that $i_k = j_k$ since this minimizes the number of gradient evaluations required per iteration. Such couplings are known as maximal couplings in the literature (see, e.g., Biswas et al. (2019)) and can easily be computed for discrete random variables. With a maximal coupling, the expected number of gradient evaluations per iteration becomes $1 + \|v - 1/n\|_{TV}$. Using this idea we arrive at SRG+ described in Algorithm 2.

Let us briefly discuss the implementation of SRG+. Sampling from $\pi$, the maximal coupling of $v$ and the uniform distribution, requires forming three probability vectors (see, e.g., Biswas et al. (2019)). As both $v$ and $1/n$ are constant throughout the optimization process, we can form these vectors at the beginning of the algorithm along with their partial sums for a total initial cost of $O(n)$ operations. We can then sample from $\pi$ in $O(\log n)$ time using binary search on the partial sums at each iteration. Adding this sampling therefore does not change the overhead of SRG.

## 5.2 ANALYSIS

The analysis of SRG+ is similar to that of SRG. In particular, the iterates of SRG+ also obey a slight modification of Theorem 1 where the bound on the largest allowable step size is loosened to $\alpha_k \leq \theta_k / 12\overline{L}$. We refer the reader to Appendix C for more details. Due to this improvement, we get that the complexity of SRG+ with a constant mixture coefficient and step size is given by:

$$O\left(n + \sqrt{\frac{n\sigma_*^2}{\mu\varepsilon}} + \overline{\kappa} + \frac{\sigma_*^2}{\mu^2\varepsilon}\right) \log\left(\frac{1}{\varepsilon}\right)$$

This shows that SRG+ performs both variance reduction as shown by the dependence of the complexity on $\sigma_*^2$ instead of $\sigma^2$ and preconditioning as shown by the dependence on $\overline{\kappa}$ instead of $\kappa_{\max}$.

## 6 RELATED WORK

At a high-level, three lines of work exist that study the use of importance sampling with SGD. The first one considers fixed importance sampling distributions based on the constants of the problem (Needell et al., 2014; Zhao & Zhang, 2015), and shows that such a strategy leads to improved conditioning of the problem. The second considers adaptive importance sampling and similar to our work targets the variance of the gradient estimator, but generally fails at providing strong convergence rate guarantees under standard assumptions (Papa et al., 2015; Alain et al., 2016; Canevet et al., 2016; Stich et al., 2017; Katharopoulos & Fleuret, 2018; Johnson & Guestrin, 2018). The third line of work frames the problem as an online learning problem with bandit feedback and provides guarantees on the regret of the proposed distributions in terms of the variance of the resulting estimators (Namkoong et al., 2017; Salehi et al., 2017; Borsos et al., 2018; 2019; El Hanchi & Stephens, 2020). Our method is closely related to the one proposed by (Papa et al., 2015); their analysis however requires non-standard assumptions and their main result is asymptotic in nature. In contrast, our work is the first to provide non-asymptotic guarantees on the suboptimality of the iterates for variance-reducing importance sampling under standard technical assumptions.

## 7 EXPERIMENTS

In this section, we empirically verify our two main claims: (i) SRG performs variance reduction which can improve the asymptotic error of SGD. (ii) SRG+ performs both variance reduction and preconditioning, and can both reduce the asymptotic error of SGD and allow the use of larger step

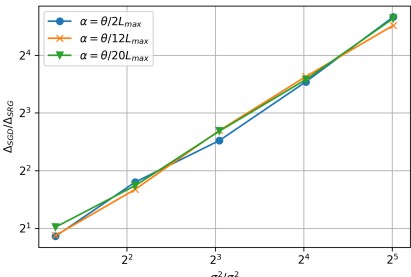 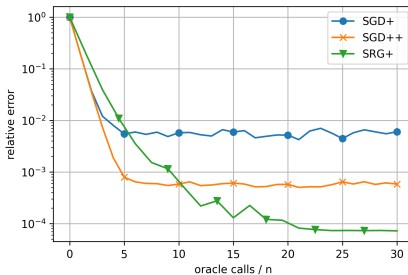

Figure 1: Left: Linear dependence of $\Delta_{SGD}/\Delta_{SRG}$, the ratio of asymptotic errors of SGD and SRG, on $\sigma^2/\sigma_*^2$, the ratio of the uniform and optimal variances at the minimum. Right: SRG+ achieves smaller asymptotic error than both SGD with $L_i$-sampling (SGD+) and SGD with partially biased sampling (SGD++), while using large $O(1/\overline{L})$ step sizes just like SGD+ and SGD++.

sizes. We start with controlled synthetic experiments that provide direct support for our claims. We then compare SRG to other baseline optimizers on $\ell_2$-regularized logistic regression problems. In all experiments, we ran each algorithm 10 times and present the averaged result.

### 7.1 SYNTHETIC EXPERIMENTS

For our first experiment, we consider the following toy problem. We let $x \in \mathbb{R}$ and $f_i(x) = \frac{1}{2}(x - a_i)^2$ where $a_i = 0$ for $i \in [n-1]$ and $a_n = 1$. In this case, $x^* = 1/n$, $\sigma^2 \approx 1/n$, and $\sigma_*^2 \approx 4/n^2$. We consider five instantiations of this problem with $n \in \{8, 16, 32, 64, 128\}$, yielding ratios $\sigma^2/\sigma_*^2$ approximately in $\{2, 4, 8, 16, 32\}$. For each instantiation, we ran SGD and SRG until they reached stationarity and recorded their asymptotic errors $\lim_{k \to \infty} \mathbb{E}\left[\|x_k - x^*\|_2^2\right]$, which we denote by $\Delta_{SGD}$ and $\Delta_{SRG}$ respectively. We experimented with three different step sizes, two of them allowed by Corollary 1, and one larger one for which we can only prove that SRG has a similar convergence guarantee as SGD. For SRG, we used the mixture coefficient $\theta = 1/2$.

We plot $\Delta_{SGD}/\Delta_{SRG}$ against $\sigma^2/\sigma_*^2$ in Figure 1 (left) for each of the three step sizes, from which we see that the relationship between the two ratios is linear, and very close to identity. From an asymptotic error point of view, these results support our theory in that the improvement is seen to be directly proportional to the ratio $\sigma^2/\sigma_*^2$. On the other hand, the constant $6(1 + 2\theta)/\rho$ in Corollary 1 would suggest that the proportionality constant is much smaller than 1, particularly when $n$ is large, but this is not what we observe in practice. This could be because the first term of the Lyapunov function (8) is quite large at stationarity, or because the constants in our bound are not sharp due to the multiple uses of Young's inequality. This latter possibility is further supported by the fact that we see a similar behaviour for SRG for step sizes larger than the ones allowed by our theory. We have consistently made these two observations in other experiments. It is however unclear to us how our analysis can be improved to match these observations.

For our second experiment, we considered the following problem. We let $x \in \mathbb{R}$ and $f_i(x) = \frac{1}{2}L_i(x - a_i)^2$, and fixed $n = 20$. Similar to the first experiment, we take $a_i = 0$ for $i \in [n-1]$ and $a_n = 1$. We then set $L_1 = n - 1$, $L_n = 1/n$, and $L_i = \frac{n-1}{n(n-2)}$, so that $\overline{L} = 1$, $L_{\max} = n - 1$. In this case, we get $x^* = 1/n^2$, $\sigma_+^2 = \sigma^2(x^*, v) \approx 1/n$, $\sigma^2 \approx 2/n^2$, and $\sigma_*^2 \approx 4/n^3$. We then ran SGD with $L_i$-sampling (SGD+), SGD with partially biased sampling (SGD++, Needell et al. (2014)), and SRG+, all with the largest allowable step size $\alpha = \theta/12\overline{L}$ from our theory for SRG+, and we use the mixture coefficient $\theta = 1/2$ for both SRG+ and SGD++. We have obtained very similar results when using the larger step size $\alpha = \theta/2\overline{L}$, which is the maximum allowable for SGD+.

We plot the relative error $\mathbb{E}\left[\|x_k - x^*\|_2^2\right] / \|x_0 - x^*\|_2^2$ for each algorithm against the number of gradient oracle calls it makes in Figure 1 (right). We see that all three algorithms are able to converge even when using the large $O(1/\overline{L})$ step sizes. The theory of SGD+ allows for larger step sizes compared to SGD, but the asymptotic error of the algorithm depends on $\sigma_+^2$ (Needell et al., 2014), which can be up to $n$ times bigger than $\sigma^2$. To solve this problem, Needell et al. (2014) proposed

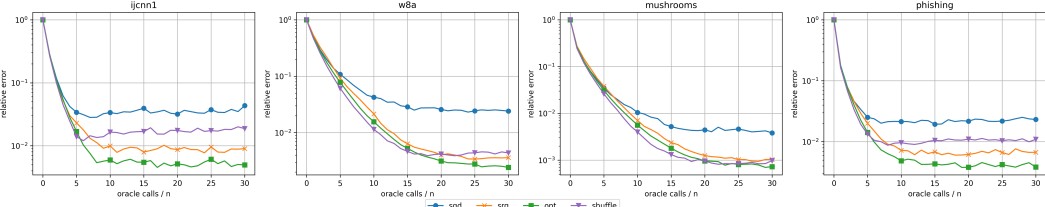

Figure 2: Comparison of the evolution of the average relative error $\|x_k - x^*\|_2^2 \, / \, \|x_0 - x^*\|_2^2$ for different optimizers on $\ell_2$-regularized logistic regression problems using the datasets *ijcnn1, w8a, mushrooms, phishing*. We compare SRG (orange) with SGD (blue), SGD with random shuffling (purple), and SGD with the optimal variance-minimizing distributions at each iteration (green).

partially biased sampling which mixes the smoothness-based distribution with uniform sampling, allowing the use of larger step sizes just like SGD+, but preserving the asymptotic error of SGD. SRG+ further improves on SGD++, and reduces the asymptotic error, making it proportional to $\sigma_*^2$.

## 7.2 $\ell_2$-REGULARIZED LOGISTIC REGRESSION

For our last experiment, we test SRG on $\ell_2$-regularized logistic regression problems. In this case, the functions $f_i$ are given by:

$$f_i(x) := \log\left(1 + \exp\left(-y_i a_i^T x\right)\right) + \frac{\mu}{2}\|x\|_2^2$$

where $y_i \in \{0, 1\}$ is the label of data point $a_i \in \mathbb{R}^d$. Each $f_i$ is convex and $L_i = 0.25\|a_i\|_2^2 + \mu$ smooth. Their average $F$ is also $\mu$-strongly convex. As is standard, we select $\mu = 1/n$.

We experiment with SRG on four datasets from LIBSVM (Chang & Lin, 2011): *ijcnn1, w8a, mushrooms* and *phishing*. For each dataset, and to be able to efficiently run our experiments, we randomly select a subset of the data of size $n = 1000$. As the datasets are normalized, we have that $\|a_i\|_2 = 1$ for all $i \in [n]$. This makes $L_i = L = 0.25 + \mu$, which reduces SGD+ to SGD, and SRG+ to SRG.

We tested the performance of SRG against three baselines, the first of which is standard SGD. The second is SGD with the optimal variance-minimizing probabilities $p_k^i \propto \|\nabla f_i(x_k)\|_2$, which allows us to compare SRG with the best possible variance-reducing importance sampling scheme. The last baseline is SGD with random shuffling, which is also known to improve the asymptotic error of SGD (under the additional assumption that each $f_i$ is also strongly convex), (Mishchenko et al., 2020). We evaluate the performance of the algorithms by tracking the average relative error $\mathbb{E}\left[\|x_k - x^*\|_2^2\right] / \|x_0 - x^*\|_2^2$. We used the mixture coefficient $\theta = 1/2$ for SRG, and used the same step size $\alpha = \theta/2L$ for all algorithms.

The results of this experiment are shown in Figure 2. We observe that SRG consistently outperforms SGD on all datasets, and that it closely matches the performance of SGD with the variance-minimizing distributions, which it tries to approximate. We also see that SRG outperforms SGD with random shuffling on two datasets, and is competitive with it on the remaining two.

## 8 CONCLUSION

We introduced SRG, a new importance-sampling based variance-reduced optimization algorithm for finite-sum problems. We analyzed its convergence rate in the strongly convex and smooth case, and showed that it can improve on the asymptotic error of SGD. We also introduced SRG+, an extension of SRG which simultaneously performs variance reduction and preconditioning through importance sampling. We expect our algorithms to be most useful in the medium accuracy regime, where the required accuracy is higher than the one achieved by SGD, but low enough that the overhead of classical variance reduced methods becomes significant. Finally, an interesting future direction would be to explore non-greedy strategies for the design of importance sampling distributions for SGD that not only minimize the variance of the current gradient estimator, but also take into account the variance of the gradient estimators of subsequent iterations.

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
