# OpenReview forum: "Stochastic Reweighted Gradient Descent"
_ICLR.cc/2022/Conference — ICLR 2022 Submitted_

### Official Review · Reviewer_N3uR · 2021-11-01

**Correctness:** 4
**Technical Novelty And Significance:** 3
**Empirical Novelty And Significance:** 3
**Recommendation:** 6
**Confidence:** 4

**Main Review:**


One of the biggest challenge in this paper is the assumption for SRG+ that we know all the Lipschitz constants of f_i s. For linear & logistic regressions, I agree that we can know them. But for many problems, especially in real world scenarios, it will be difficult to either compute these values or use them to do sampling. More comments on this is needed in the paper.

I would like to see more applications and experiments than those in the paper. The analysis does not convince me that importance sampling is truly helpful. Does this matter? Should I try to use this? Do I have any hope of using it in settings that are large-scale enough that I want to simply stream the data?

Since there is extra computation at each iteration compared to SGD (computation of norms, updates....), then experiments with run time in x-axis make more sense?

What about minibatch versions of your methods, do they support this? how? comments on this are needed.

What is the sensitivity of your methods to theta?

**Summary Of The Paper:**

This work proposes a variance-reduced method called stochastic reweighted gradient (SRG) method which is based on importance sampling.  SRG improves on the of stochastic gradient descent error for the strongly convex and smooth objective functions. The authors extended  also SRG to SRG+ to combine the importance-sampling-based preconditioning and variance reduction. Numerical experiments are provided to assess the numerical efficiency of these methods.

**Summary Of The Review:**

See above

---

> ### Author Response · Authors · 2021-11-19
> **Our main method does not require knowledge of the smoothness constants.**
>
> \> *"One of the biggest challenges in this paper is the assumption for SRG+ that we know all the Lipschitz constants of f_i s."*
>
> The main algorithm in the paper (SRG) does not require knowledge of the smoothness constants. SRG+ simply says that IF one has access to these constants, then one can leverage this knowledge to design better importance sampling distributions. If such knowledge is not available, then one can naturally fall back to SRG.

---

> ### Author Response · Authors · 2021-11-19
> **We added a mini-batch version of SRG.**
>
> \> *"What about minibatch versions of your methods [?]"*
>
> We added a mini-batch version of SRG in the appendix.

---

> ### Author Response · Authors · 2021-11-19
> **Are there any specific experiments that you believe would significantly contribute to supporting the claims of the paper ?**
>
> \> *"I would like to see more applications and experiments than those in the paper."*
>
> Are there any specific experiments that you believe would significantly contribute to supporting the claims of the paper ? We'd be happy to consider running them if they strengthen our contribution. If referring to NN experiments, please see our comment above titled "Limitedness of setting and extension to neural networks.".

---

> ### Author Response · Authors · 2021-11-19
> **SRG introduces negligible overhead compared to SGD.**
>
> *"Since there is extra computation at each iteration compared to SGD (computation of norms, updates....), then experiments with run time in x-axis make more sense?"*
>
> The extra-computation needed by SRG is negligible. We discuss in the paper how SRG can be implemented using only an additional $O(\log{n})$ floating point operations per iteration when compared to SGD. We did not compare runtime because our implementation was a non-optimized research implementation.

---

> ### Author Response · Authors · 2021-11-19
> **SRG should be used whenever possible.**
>
> \> *"Does this matter? Should I try to use this?"*
>
> Up to small constants, our analysis asserts that SRG will perform better than SGD when $\sigma_*^2 \ll \sigma^2$. This is further supported by our experiments, particularly Figure 1 (left). Because $\sigma_*^2 \leq \sigma^2$ always holds, and because the overhead of SRG is negligible, there is no reason not to use SRG instead of SGD.

---

> ### Author Response · Authors · 2021-11-19
> **SRG is not too sensitive to $\theta$ as long as it is reasonably large.**
>
> \> *"What is the sensitivity of your methods to theta?"*
>
> The short answer is that our method is not too sensitive to $\theta$ as long as it is reasonably large. A good default when using a constant step size is to set $\theta = 1/2$ as we did in our experiments.
>
> Our theoretical results provide a more precise answer to this question (Theorem 1). The choice of theta controls a trade-off: a smaller $\theta$ leads to a smaller asymptotic error, while a larger theta allows the use of a larger step size leading to a faster initial convergence.
>
> We view $\theta$ as a parameter that the user can set depending on their goal (rather than a hyperparameter which should be optimized over). Our suggestion would be to start the algorithm with the largest theta allowable ($\theta$ = 1/2), then decrease it (jointly with the step size, as prescribed by our theory, see Theorem 1) as needed to achieve higher accuracy.

---

### Official Review · Reviewer_wvfK · 2021-11-02

**Correctness:** 4
**Technical Novelty And Significance:** 2
**Empirical Novelty And Significance:** 2
**Recommendation:** 5
**Confidence:** 4

**Main Review:**

1- The paper is well-written and it is easy to follow and walk through the proofs.

2- The major comment is that the analysis is for very limiting settings. It would be more practical to analyze the method for non-convex settings to see the effectiveness of the method in practice for training NNs. It would be useful to apply this method to NN training.

3- For the experiments, it is not clear how many times each experiment has been run and if the result is the average of those.



**Summary Of The Paper:**

This paper considers minimizing the finite sum problem with SGD with non-uniform sampling.
The optimal sampling weight at a given point is achieved via minimizing the variance of stochastic gradient which needs the norm of the gradient of each function at that point. To approximate these sampling weights and also be efficient this paper proposes to take the approach similar to SAGA and use auxiliary memory with the size of number of functions and update one of the stored values in each iterate similar to SAGA. Using stale and stored gradient norms gives an upper bound to the stochastic gradient variance that could be minimized similarly. Since this new upper bound contains two terms, they propose a mixture of non-uniform and uniform sampling to make both terms small. They analyze SGD with the proposed sampling scheme and show that the sub-optimality upper bound is n (the number of functions)-times smaller than the upper bound of SGD with uniform sampling. They also propose another weighting scheme that mixes uniform and non-uniformes schemes proportioned with gradient norms and smoothness parameters of each function.


**Summary Of The Review:**

The major comment is that the analysis is for very limiting settings. It would be more practical to analyze the method for non-convex settings to see the effectiveness of the method in practice for training NNs. It would be useful to apply this method to NN training.

---

> ### Author Response · Authors · 2021-11-19
> **Limitedness of setting and extension to neural networks.**
>
> \> *"The major comment is that the analysis is for very limiting settings."*
>
> While we agree that our setting (strongly-convex and smooth objectives) is somewhat limited, it is the standard setting under which new optimization algorithms are analyzed. Let us also point out that there is a large literature on importance sampling for variance reduction for SGD, yet our paper is the first that provides a clean analysis of such methods under the standard technical assumptions of smoothness and strong-convexity. We believe that this is significant and useful to the community on its own.
>
> \> *"It would be more practical to analyze the method for non-convex settings to see the effectiveness of the method in practice for training NNs. It would be useful to apply this method to NN training."*
>
> We agree that an extension of our results to the non-convex case would be nice and possibly within reach using ideas in [1], but we left that to future work. In particular, It is not clear to us that such an analysis would uncover the practical behaviour of SRG on the training of NN. Experiments would be the most informative.
>
> However, our method is simply not efficiently implementable within current deep learning frameworks for large NNs that are used in practice. In particular, at each iteration, one would need to obtain the per-example gradients within each mini-batch to be able to calculate their norms. In current implementations, this is infeasible for all but the smallest NN architectures (holding such per-example gradients in working memory is not feasible, even for relatively small batch sizes).
>
> The question of whether per-example gradient norms can be efficiently computed is a research question that is outside the scope of our work. We believe this to be possible for linear layers for example where the per-example gradients can be expressed as outer products. If you believe that the above explanation is useful for readers of the paper, we would happily include it in a new section of the appendix.
>
> [1] Reddi, S.J., Sra, S., Póczos, B., & Smola, A. (2016). Fast Incremental Method for Nonconvex Optimization. ArXiv, abs/1603.06159.

---

> ### Author Response · Authors · 2021-11-19
> **Each experiment was run 10 times and the average of these runs is what is displayed.**
>
> \> *"For the experiments, it is not clear how many times each experiment has been run and if the result is the average of those."*
>
> Each experiment was run 10 times and we display the average of the 10 runs. We have added this detail to our paper.

---

### Official Review · Reviewer_i1ZH · 2021-11-02

**Correctness:** 3
**Technical Novelty And Significance:** 3
**Empirical Novelty And Significance:** 3
**Recommendation:** 5
**Confidence:** 3

**Main Review:**

There is a rich literature in the domain for SGD based on importance sampling and conditional variances approximation. The paper combines these two ideas, and I believe that it is useful for reducing the asymptotic error and computational complexities. I have some concerns as follows.
1.	I might be misunderstanding part of the proofs, but I hope the author could help me understand more on why Lemma 2 holds. In the algorithm, $p_k$ is calculated based on $q_k$, which is related to a binomial distribution with parameter $\theta_k$. So the randomness of $q_k$ may have an impact on the expectation. However in the proof, the impact of $q_k$ is not involved, and $p_k$ seems to be treated as a constant. In other words, since the importance sampling process can be regarded as a kind of Bayes procedure, how does the distribution of the sampling probability impact the convergence?
2.	In the experiments, the proposed method is compared with SGD+ (SGD with $L_i$-sampling) and SGD++ (SGD with partially biased sampling), how about comparing with importance sampling SGD with the optimal sample probabilities?


**Summary Of The Paper:**

The authors propose the stochastic reweighted gradient (SRG) algorithm, which is based on importance sampling in SGD to reduce the variance for the SGD method. The authors also extend SRG to combine the benefits of both importance-sampling-based preconditioning and variance reduction. The convergence rate is studied and the authors show that the proposed method can achieve a better asymptotic error than SGD.

**Summary Of The Review:**

I believe the paper has its value in combining importance sampling SGD with conditional variances approximation to reduce the asymptotic error as well as the computational complexity. I just have one question on the technical soundness of the convergence results.

---

> ### Author Response · Authors · 2021-11-19
> **$p_k$ is constant conditional on $(i_t, b_t)_{t=0}^{k-1}$**
>
> \> *"[...] the randomness of $q_k$ may have an impact on the expectation. However in the proof, the impact of $q_k$ is not involved, and $p_k$ seems to be treated as a constant [...] how does the distribution of the sampling probability impact the convergence?"*
>
> Thank you for the question. The randomness of $p_k$ (due to the randomness of $q_k$) does indeed have an impact on the *overall* expectation. The reason this does not interact directly with our proof is because our bound is on the *conditional* expectation of the second moment, conditioned on $(b_t, i_t)_{t=0}^{k-1}$. Conditioned on these random variables, $q_k$ is constant and so is $p_k$, and we can proceed with the analysis.

---

> ### Author Response · Authors · 2021-11-19
> **We do compare SRG with the optimal importance sampling probabilities.**
>
> \> *[...] how about comparing with importance sampling SGD with the optimal sample probabilities ?*
>
> We compare SRG and SGD with the optimal importance sampling probabilities in the experiments depicted in Figure 2. We did not include this comparison in the experiments of Figure 1 since our goal there was to verify our theoretical claims rather than compare our methods to baselines.

---

### Official Review · Reviewer_F7eT · 2021-11-03

**Correctness:** 4
**Technical Novelty And Significance:** 2
**Empirical Novelty And Significance:** 2
**Recommendation:** 6
**Confidence:** 4

**Main Review:**

The importance sampling procedure is new, however there have been many similar works in the past and this paper's results maybe very incremental.

**Summary Of The Paper:**

This paper uses importance sampling in SGD to improve performance.

**Summary Of The Review:**

The importance sampling procedure is new, however there have been many similar works in the past and this paper's results maybe very incremental.

---

> ### Author Response · Authors · 2021-11-19
> **Can you defend your claim that our paper is very incremental ?**
>
> We laid out our contributions in the introduction and the relationship between our method and analysis to the literature in our related work section. Can you defend your claim that our paper is very incremental ?

---

### Comment · Reviewer_N3uR · 2021-11-29
**After rebuttal**

I have read the rebuttals. The authors answered "correctly" some of my questions  and responded with general statements to others. I also read the other criticisms from other reviews. I still think this work has some merit as it is, but I cannot increase my score to more than 6.

---

### Decision · Program_Chairs · 2022-01-20

**Decision:**

Reject

**Comment:**

The reviewers and AC all find the presented approach interesting and promising.

However, as pointed out in the reviews and as the authors recognized, the strongly convex + smooth objective setting considered is limited. Given the prevalence of non-convex settings in many practical applications and the rich related literature on the analysis of SGD and variants in the non-convex setting,  it would be highly desirable to (i) consider experiments on small NN architectures (since the method cannot accommodate larger architectures)  to gain some understanding of the value of the approach and (ii)  to try and extend the present analysis to the non-convex case.

It would also be valuable to perform experiments illustrating the impact of theta indicated by the theory.